# Interpersonal Competition in Elderly Couples: A Functional Near-Infrared Spectroscopy Hyperscanning Study

**DOI:** 10.3390/brainsci13040600

**Published:** 2023-03-31

**Authors:** Qian Zhang, Zhennan Liu, Haoyue Qian, Yinying Hu, Xiangping Gao

**Affiliations:** 1Department of Psychology, Education College, Shanghai Normal University, Shanghai 200234, China; 1000481182@smail.shnu.edu.cn (Q.Z.);; 2Institute of Neuroscience, Key Laboratory of Primate Neurobiology, CAS Center for Excellence in Brain Science and Intelligence Technology, Chinese Academy of Sciences, Shanghai 200031, China; 3Shanghai Institute of Early Childhood Education, Shanghai Normal University, Shanghai 200234, China

**Keywords:** elderly couples, competition, hyperscanning, fNIRS

## Abstract

Elderly people tend not to compete with others, and if they do, the mechanism behind the competition is not clear. In this study, groups of elderly couples and matched cross-sex controls were recruited to perform a competitive button-pressing task, while their brain signals were simultaneously collected using functional near-infrared spectroscopy (fNIRS) hyperscanning. Several fundamental observations were made. First, controls showed attenuated interpersonal competition across task processes, but couples held the competition with each other. Second, couples demonstrated increased inter-brain synchronization (IBS) between the middle temporal cortex and the temporoparietal junction across task processes. Third, Granger causality analysis in couples revealed significant differences between the directions (i.e., from men to women, and from women to men) in the first half of the competitive task, whereas there was no significant difference in the second half. Finally, the groups of couples and controls could be successfully discriminated against based on IBS by using a machine-learning approach. In sum, these findings indicate that elderly couples can maintain interpersonal competition, and such maintenance might be associated with changes in the IBS of the mentalizing system. It suggests the possible positive impact of long-term spouse relationships on interpersonal interactions, both behaviorally and neurally, in terms of competition.

## 1. Introduction

People’s pro-sociality grows with age [1,2,3,4,5], while the preference for competition declines in the older years [6]. For older people, intimate relationships affect their pro-sociality [7,8,9,10,11,12], and couples usually exhibit more pro-sociality with each other [13,14]. However, the influence of intimate relationships on competition remains unknown. In this study, we investigated the competitive interaction within elderly couples and the inter-brain synchronization (IBS) during their competition.

For elders, the relationship with his/her spouse is the principal intimate relationship in their later life [15], and such a relationship influences their cognitive and social activities. For example, elderly people living with a spouse have better cognitive abilities than those living alone or with offspring [16]. Moreover, the loss of a spouse has negative effects on the elderly’s cognitive functions (e.g., working memory), accompanied by more social withdrawal [17]. Another study reported that elderlies without a spouse are more vulnerable to depression in social life relative to those with a spouse [18]. In addition, there is empirical evidence that elderly couples have similarities in cognitive functions (e.g., lexicosemantic) because of the mutual influence between them [19]. Elderly couples also experience more enjoyment and less anxiety during social interactions with their spouse [20,21]. Together, these findings suggest that elderly couples might have greater competition when they interact with each other.

As one basic form of social interaction, interpersonal competition has benefits in inspiring effort and enjoyment [22,23,24,25], which also works well for the elders [26]. Previous studies have shown the complex effects of intimate relationships on interpersonal competition. Some researchers revealed that compared to opposing strangers, competing with friends evoked smaller P300 amplitudes in the frontoparietal cortex [27], which negatively correlates with empathy to another’s personal distress [28]. In contrast, other researchers found that playing against a friend (vs. a stranger) elicited higher engagement and physical arousal during a video game [29]. It has been reported that there were increased activations in the temporoparietal junction (TPJ) when competing with familiar friends (vs. strangers), which positively correlated with social motivation [30]. Given that there is more enjoyment and less anxiety between elder couples, we hypothesized greater competition within elder couples when they compete with each other.

Previous neuroimaging studies have shown increased brain activity during interpersonal competition within a single brain in the prefrontal cortex (PFC) and TPJ [31,32,33]. To investigate the nature of interpersonal competition (i.e., a feedback loop in a multi-brain system), a growing number of studies have adopted the technology of hyperscanning (i.e., simultaneously recording brain activities from two or more individuals). During the interpersonal competition, there was enhanced inter-brain synchronization (IBS) in the frontal cortex, the parietal cortex, and the temporal cortex [34,35,36,37]. Further, IBS during interpersonal competition was found to be associated with behavioral performance, that is, higher IBS, more intense competition [38,39]. Thus, these results suggest that the IBS can be used as a neural marker of interpersonal competition.

Besides interpersonal competition, IBS has been observed in various social interactions, such as joint action [40], eye-to-eye contact [41], verbal communication [42], and interpersonal touch [43]. Previous hyperscanning studies have shown that IBS during social interactions is affected by intimate relationships between interacting individuals [44,45]. For instance, long-term couples showed the highest IBS in sensorimotor areas during motor coordination compared to best friends and unfamiliar group members [46]. It is also reported that there is enhanced IBS during social interactions in romantic lovers [47] and parent–child dyads [48,49]. The enhanced IBS has been mapped in the frontal cortex and TPJ [50,51,52]. Therefore, we also hypothesized enhanced IBS in elderly couples during interpersonal competition.

The present study aimed to explore the interpersonal competition between elder couples through the fNIRS-based hyperscanning method. This method is thought to be more suitable for collecting brain signals in social interactions as well as in older people compared with EEG and fMRI [53]. Elderly couples and matched cross-sex control couples were asked to perform a competition task in which they had to respond (i.e., pressing the key to a stimulus) faster than their partner [54,55]. The brain signals of pairs of participants were simultaneously recorded while they were performing the competition task. Because of the localization of the IBS during interpersonal competition and human attachment, bilateral PFC and TPJ are selected as the region of interest (ROI). We anticipated that there would be better behavioral performance and increased IBS in elderly couples than in matched controls during the competition task. Specifically, we hypothesized that the competition would be greater or more durable with increased IBS within elderly couples due to their long-term spouse relationship. Previous studies have shown that elderly couples perform better in forms of other interactions, such as cooperation [56], yet it remains unclear how their performance would be in competitions.

To our knowledge, this is the first study that focuses on the competition within elderly couples and their inter-brain synchronization. As stated at the beginning, elderly people generally avoid competition [6]. The present study aims to investigate whether there would be any difference within a long-term spouse relationship. Specifically, would the elderly person’s competition be longer and with greater brain synchronization if the competitor is their spouse?

## 2. Materials and Methods

### 2.1. Participants

Twenty-two elderly couples arrived at the laboratory, and data from five of them were deleted since the dyads failed to finish the task or for technical reasons. The remaining were seventeen elderly couples (age range: 62–78 years, mean age ± SD = 67.44 ± 5.15 years, 17 females) who completed the whole task. All couples were in their first marriage, which had lasted for at least thirty years. Thirty-six elders (age range: 58–80 years, 65.94 ± 5.05 years, 18 females) were recruited as controls. They were randomly paired into 18 cross-sex dyads. The two members of one dyad in the controls were unacquainted with each other prior to the experiment. All participants were right-handed, with normal or corrected-to-normal vision. Participants were required to complete a questionnaire about personal information and healthy states upon arriving at the laboratory. None of them reported any history of psychiatric or neurological diagnoses, including depression. Moreover, the Montreal Cognitive Assessment (MoCA) was conducted individually before the experiment in order to evaluate their cognitive functions, and all participants had a score of 26 or more. Each participant signed written informed consent prior to the experiment and was paid for his or her participation. The research was approved by the ethics committee of Shanghai Normal University.

### 2.2. Tasks and Procedures

Participants were instructed to complete a computer-based interpersonal competition task. In total, the task included 40 trials (Figure 1A). Before the formal task, participants were asked to practice for several trials to ensure that they fully understood the task. In the formal experiment, each dyad of participants sat side-by-side in front of a shared computer screen, with verbal or non-verbal communication not allowed between each other throughout the experiment. The competition task was comprised of rest period 1 (30 s), 20 trials of task block 1 (~150 s), rest period 2 (30 s), 20 trials of task block 2 (~150 s), and rest period 3 (30 s). Each trial began with a hollow gray circle (0.6~1.5 s) presented in the center of the screen. The hollow circle was then filled with green, signaling participants to press the key (i.e., button ‘0′ on a keyboard for the player who sat on the right side, and button ‘1′ for the player who sat on the left side) using their right index fingers. Dyads of participants had to respond as fast as possible to compete with each other. They were told that if they responded faster than their opponent, they would win two points, but they would lose two points and vice versa. Once both participants of dyads had responded, one feedback was presented (4 s). The feedback contained the following information: (i) who was faster or slower, (ii) the points each one won or lost in this trial, and (iii) the cumulative points for each participant in the task. Finally, a blank screen appeared for 2 s.

### 2.3. fNIRS Data Acquisition

The NIRS system (NIRScout, NIRx Medical Technologies, LLC, Berlin, Germany) was used to record the brain signals of two members of one dyad during the whole experiment. The changes in blood oxygenation at two wavelengths (760 and 830 nm) were measured with a sampling rate of 7.81 Hz. For each participant, three sets of optode probes were placed on the head to cover bilateral PFC and TPJ, with a total of 23 recording channels (Figure 1B). For each probe set, the distance between emitters and detectors was about 3 cm. The placements of the emitter and detector were referred to as the 10–20 system [57]. Specifically, the center of the probe set of PFC was placed at the Fpz position; and the center of the probe set of TPJ was pointed at the C5 and C6 positions (i.e., C5 for the left TPJ, C6 for the right TPJ). The brain region of each channel was obtained by registering to a Montreal Neurological Institute (MNI) standard brain template using the SPM8 toolbox [58].

### 2.4. Data Analysis

#### 2.4.1. Behavioral Data Analysis

The response time (RT) of each trial was recorded for each participant. First, the RTs that were longer than 1500 ms (less than 1%) were excluded to eliminate the effects of outliers [59]. Next, the mean values and standard deviations were calculated for each participant. RTs were further excluded when they were over the mean ± 3 SD. Finally, the performance of competition was accessed by computing the difference in response times (DRT) between two participants of one dyad, with a smaller DRT indicating more intensified competition.

In order to compare the performance between groups, the covariance analysis (ANCOVA) was performed on the DRT with a group (couple, control) as a between-subject factor and a task block (block 1, block 2) as a within-subject factor. Meanwhile, the age of men and women, as well as other demographic variables (e.g., who handled the money in couples), were included as covariates in ANCOVA.

#### 2.4.2. fNIRS Data Analysis

The fNIRS data were preprocessed through Homer3 [60]. Raw data were visually inspected, and coefficients of variation (i.e., CV) [61,62] were used in channel exclusion. Specifically, channels with CV > 7.5% were defined as bad channels and excluded from further analysis since they may contain unphysiological noise [61,62]. Subjects with over 30% bad channels were defined as failing the experiment and were thus deleted. The raw optical data were converted into optical density units. Then, the discrete wavelet transformation filter was used to detect and correct motion artifacts [63]. The principal component analysis was also used to remove global physiological noise (i.e., skin blood flow), while 80% of the covariance of the data was removed [64]. Subsequently, the data were down-sampled to 1 Hz. Finally, the modified Beer–Lambert law was adopted to obtain the HbO and HbR concentrations.

Since HbO was sensitive to the changes in the regional cerebral blood flow [65], we only included the processed HbO concentrations in the subsequent analysis. In order to assess the IBS during interpersonal competition, the wavelet transform coherence (WTC) method was used on the processed HbO [66]. WTC is the most widely used approach to assess IBS in fNIRS hyperscanning studies [51,52,54,67], which could evaluate the cross-correlation between two movement signals on the time-frequency plane. In addition, WTC is effective in removing low-frequency noise as hemodynamic signals are transformed into wavelet components.

To determine the task-related frequency band, we conducted statistical tests on the IBS difference (i.e., task vs. baseline) across the full frequency range (e.g., 0.01–1 Hz) [45,68,69,70]. In this way, we estimated IBS by using WTC for the task period (i.e., ~300 s) and the resting period (i.e., 90 s) for each dyad. The resting state here was used as the baseline. Then, the IBS for the task and baseline were both converted to Z-values, respectively. A series of t-tests were conducted to compare the converted IBS during the task with respect to the baseline. All possible combinations for channels between the two members of one dyad were examined (i.e., 23 × 23 = 529 in total). The multiple-comparison problem was corrected by using the false discovery rate (FDR) method (*p* < 0.005). The results showed enhanced IBS for frequencies ranging between 0.11 and 0.14 Hz. This frequency band was in accordance with previous studies that used the same task (e.g., [52,54,55]), as well as the period of one trial in our experiment task (i.e., 0.10~0.17 Hz). Within the selected frequency band, the IBS was averaged across groups (i.e., couples, controls) and task blocks (i.e., block 1, block 2), and compared using ANCOVA.

To further verify that the identified IBS was specific to interpersonal competition, a validation method of the permutation test was conducted. Elderly couples were randomly assigned to form new dyads that did not actually compete with each other, and then the IBS was recalculated across channels. The ANCOVA was used to compare the IBS in different groups and task blocks. This permutation test was repeated 1000 times to generate a distribution (F value) of all channels and compared to the real data. The results were corrected by using the FDR method (*p* < 0.05).

In addition, a time-lag analysis on IBS was performed since a previous hyperscanning study showed that IBS usually involved a time lag in couples [45]. In order to test the possible lead-lag pattern between men and women of couples, various time-lags were added when computing the IBS. Specifically, we shifted men’s time course forward or backward relative to those of women from 2 to 10 s (step = 2 s), with other computations similar to the latter part of the above paragraph.

Granger causality analysis (GCA) was performed in order to evaluate the direction of synchronization for channels that displayed significant IBS [71]. The causal relationship between time series in brain data was measured through vector autoregressive models used in GCA. In the present study, GCA was based on PCA-corrected signals during the task periods. We calculated the pairwise conditional G-causality of both participant directions, from men to women and from women to men. The order of the AR model was determined to be two, based on the Bayesian information criterion [72]. Then the repeated-measures ANOVA was used to test whether two directions were different in couples across task blocks.

Finally, to explore which extent the two groups could be differentiated from each other based on the demonstrated IBS, the k-nearest-neighbor (KNN) classifier was used, which was based on IBS matrices with odd k-values from 1 to 31 [73]. The leave-one-out cross-validation approach to identify the IBS difference (i.e., block 2—block 1) worked significantly better in discriminating participant groups than the mean of 10,000 random classifications. Due to the noise-sensitivity of KNN classifiers, we used the average classification accuracy of all k-values to show the time points of successful classification, which was unaffected by the option of the k-value. The areas were further required to display significantly above-chance classification with at least half of the selected k-values.

## 3. Results

### 3.1. Behavioral Performance

The ANCOVA revealed non-significant main effects of the group and task block (*F*s < 1.17, *p*s > 0.28, $\eta_{p}^{2}s$ < 0.04). The interaction effect between the group and task block was significant (*F* = 4.85, *p* = 0.04, $\eta_{p}^{2}=$ 0.14). The simple effect analysis revealed that controls had higher DRT (103.26 ± 40.23) in block 2 than that in block 1 (73.21 ± 27.00; *t* = 2.61, *p* = 0.01, Cohen’s *d* = 2.34; Figure 2). However, there was no significant difference between block 1 and block 2 for couples (87.02 ± 48.37 vs. 96.90 ± 61.14; *t* = 1.45, *p* = 0.16, Cohen’s *d* = 0.82). These results indicated that controls showed a decreased competition across task blocks, but couples could hold the competition with each other.

### 3.2. Inter-Brain Synchronization

The ANCOVA results showed a significant interaction effect between group and task block at several pairs of channels before FDR correction, that was, channel 3 of men and channel 6 of women, channel 7 of men and channel 6 of women, channel 12 of men and channel 11 of women, channel 12 of men and channel 23 of women (*F*s > 8.24, *p*s < 0.006). However, after FDR correction, only the interaction effect between the left middle temporal cortex (MTC, channel 3) of men and the right TPJ (channel 6) of women survived (Figure 3A). The simple effect analysis revealed increased MTC-TPJ IBS for couples in block 2 compared to block 1 (0.30 ± 0.08 vs. 0.38 ± 0.09; *t* = 2.46, *p* = 0.02, Cohen’s *d* = 0.61), but decreased MTC-TPJ IBS for controls from block 1 to block 2 (0.35 ± 0.10 vs. 0.29 ± 0.10; *t* = −2.14, *p* = 0.04, Cohen’s *d* = 0.51; Figure 3B). In block 2, there was also enhanced IBS in couples than in controls (*t* = 2.91, *p* = 0.01, Cohen’s *d* = 0.75). Additionally, the interaction effect at MTC-TPJ was also found when the brain activity of women lagged behind that of men by 2~8 s and when the brain activity of men lagged behind that of women by 2 s (Figure 3C).

Compared with the distribution generated by the permutation procedure, the interaction effect between group and task block at the MTC and TPJ (i.e., channel 3 of men, and channel 6 of women) occurred only in two members of a dyad who tried to compete with each other, and not in two members of recombined random dyads (Figure 3D).

### 3.3. Directional Coupling

The GCA was conducted to determine the direction of IBS during competition in couples. The results showed that the main effects of group and task block were not significant (*F*s < 2.20, *p*s > 0.15, $\eta_{p}^{2}s$ < 0.13). However, the interaction effect between the group and task block was significant (*F* = 4.75, *p* = 0.045, $\eta_{p}^{2}=$ 0.23). Further analysis revealed that in block 1, the mean G-causality was significantly larger from men to women (0.10 ± 0.05) than that from women to men (0.05 ± 0.03; *t* = −2.87, *p* = 0.01, Cohen’s *d* = 0.59; Figure 4). Meanwhile, no significant difference was found between the two directions in block 2 (0.07 ± 0.05 vs. 0.07 ± 0.07; *t* = 0.02, *p* = 0.99, Cohen’s *d* = 0.004).

### 3.4. Classification Results

We examined the extent to which one can discriminate between couple groups and control groups based on the IBS alone. We extracted the IBS difference (block 2—block 1) from the channel combinations that showed significant interaction effect between group and task block before FDR correction (i.e., channel 3 of men and channel 6 of women, channel 7 of men and channel 6 of women, channel 12 of men and channel 11 of women, channel 12 of men and channel 23 of women). Based on the four IBS differences, the KNN classifier effectively discriminated couples and controls with a classification accuracy of 87.30%, which is a significant above-chance accuracy (*p* = 2.20 × 10^−16^; Figure 5).

## 4. Discussion

The current work explored interpersonal competition in elderly couples. Behavioral results showed that response time differences in controls increased across blocks while maintained between couples, indicating better competition between couples. Moreover, fNIRS results showed increased IBS between couples across blocks. In contrast, the revealed IBS significantly decreased in controls across blocks.

The behavioral performance during interpersonal competition differed between couples and controls. Controls showed decreased competition over time, which is consistent with previous studies that elderly people are not interested in interpersonal competition [74]. However, some researchers found that elderly adults were still willing to compete with each other [75,76]. This appears to be inconsistent with our findings. However, it should be noted that participants were motivated to compete for rewards in the aforementioned studies, while participants were instructed to compete with a real person in this study. Future studies can further examine interpersonal competition in older people with or without rewards. In addition, since a button-pressing task was used in the present study, which calls for motor control abilities and reflects the influence of life habits, it would be better if these two factors could be included in the control group.

Couples held interpersonal competition, which is in line with previous findings that there is greater competition when competing with intimates [27,28]. It has been reported that competing against one’s friend is associated with better engagement and greater physiological arousal in video game competitions [29]. Moreover, competition with friends leads to higher activations in the right TPJ, striatum, and amygdala, which are correlated with subjective scores of motivation and pleasantness [30]. In the current work, the better behavioral performance between elderly couples might be due to the greater motivation of competition in intimates.

The revealed IBS in couples echoes recent findings, in which there is increased IBS between intimates when they are interacting with each other [51,77,78]. For example, romantic lovers showed IBS increases during interpersonal touch [45], interpersonal conflict [44], cooperative key-pressing [52,67], and cooperative problem-solving [47]. Further, the revealed IBS was approximately located in the middle temporal cortex and TPJ. The IBS in these areas has been interpreted as a marker for mutual understanding in previous hyperscanning studies [79,80,81]. On the former, it has been reported that there is significant and consistent IBS in MTG when two persons make eye contact [82], exchange semantic words alternatively [83], and move fingers synchronously [84]. For the IBS in TPJ, it has been observed in a nine-person drumming task [85], a naturalistic cooperation task [50], and an online cooperation task [86]. Furthermore, these brain regions have been thought to play an important role in interpersonal competition [87,88,89]. They are associated with self–other monitoring, that is, understanding and predicting the opponent’s action and, in turn, adjusting one’s own action. Accordingly, the IBS between MTG and TPJ here might mean that the elderly couples were mutually engaged in understanding and predicting the competitor’s mind so that the competition between them was maintained.

However, it should be mentioned that an alternative explanation exists for the results above. Although the 0.05–0.1 Hz frequency band was excluded from our analysis, our frequency range is close to Mayer waves, which are cyclic changes or waves in arterial blood pressure brought about by oscillations in baroreceptor and chemoreceptor reflex control systems. Therefore, one possible explanation is, stranger dyads felt stressed in the first task block, and this stress was measured during the experiment via parasympathetic activity via the Mayer waves. In order to exclude this alternative explanation, Mayer waves should be fully avoided in future research.

The classification results indicated that the IBS between MTG and TPJ was able to distinguish couples and controls in our work. These findings are in accordance with the results of previous studies that IBS is a neural marker for social interactions, in which IBS can be used to classify and predict mental processing [80,90,91]. In this work, we presented an approach of machine learning (i.e., KNN-based classifier) to reveal how well the two groups could be separated from each other based on the significant IBS. Such a method is promising in the field of interpersonal neuroscience, which provides neuroimaging researchers with a novel strategy for analyzing and validating IBS [92,93]. It would be interesting to develop comprehensive deep learning model training on the IBS during social interactions, promoting the accurate assessment of human mental processes in social interactions.

Our GCA results for couples further showed that the direction of IBS was stronger from men to women (vs. from women to men) in block 1, while the two directions were insignificantly different in block 2. Previous studies revealed significant IBS directional differences in interacting partners, such as from the woman to the man of one romantic lover-dyad in a cooperative key-pressing task [52], from the instructor to the learner in a socially interactive learning task [42,94], from the leader to the follower in a dynamic cooperation task [95,96], and from the audience to the violinist in a series of violin performance [97]. Here, the significant result in block 1 implied that the primary information flow was from men to women in elderly couples when they began to compete with each other. This finding is consistent with the view that men are generally more willing to compete than women [98,99,100]. Besides, several lines of evidence have revealed that there is no gender/role difference in the IBS direction [47,101], indicating that social interactions are not always dominated by men or women. Sometimes they are equally important. Our finding in block 2 supports this point. This is further supported by our time lag analysis, in which significant IBS was observed whenever shifting men’s brain activity forward or backward.

## 5. Conclusions

In conclusion, this study found that elderly couples also competed with each other and showed increased IBS across the competitive task progress. The direction of the IBS suggested different patterns of dominance between elderly couples across the course of interpersonal competition. This extends our understanding of the synchronized brain activities for older couples’ interactions. Further, the results of the present study suggested the positive effect of long-term spouse relationships on elderly people’s interpersonal interactions in terms of competition and their brain synchronizations. This potentially provides insights into human aging and intimate relationships and may shed light on how to deal with elderly people’s tendency to withdraw from social activities since they do not withdraw so much when their spouse is acting as their partner.

## 6. Limitations and Future Research

Our study has several limitations. First, although distinct competition patterns of IBS between couples and controls were observed in our sample, it should be further confirmed in a large sample. It has been reported that a high sample size would be required to detect significant IBS through power analysis [102]. Second, to better reach a conclusion, more controls are required. Despite the fact that matched-cross-sex elderly were used as a control group, whose depression and overall cognitive functions were assessed and controlled, it would be better if the control group further included factors such as life habits or motor control abilities, especially in consideration of the competitive button-pressing task. Third, as discussed above, the Mayer waves may not be fully avoided in the frequency band, thus allowing other possible explanations for the results. Future research should choose a frequency band that could fully avoid Mayer waves. Fourth, the spatial resolution of fNIRS (i.e., about 1~3 cm) [65] makes it difficult to record signals in the deep areas of the brain, which might also be involved in competition for elderly couples in our work. For example, there is empirical evidence that the left precuneus is also involved in dynamic strategic competition [37]. Future work should investigate the IBS during elderly couples’ competition by adopting other neural technologies with a high spatial resolution (e.g., fMRI hyperscanning). Finally, the competition task used here is simple and lacks variety. Previous studies have shown that competition contains different types, such as strategic and non-strategic competition [103]. Future work should examine the behavioral performance and IBS in elderly couples during interpersonal competition with various kinds of competition.

## Figures and Tables

**Figure 1 brainsci-13-00600-f001:**
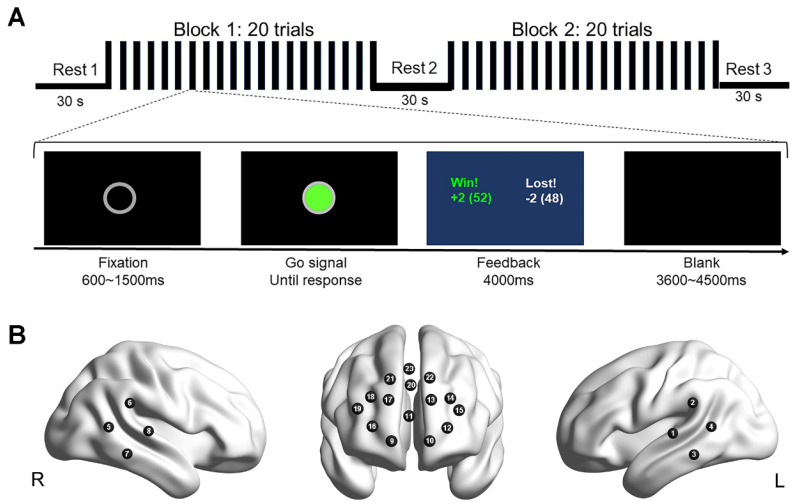
Experimental design. (**A**) Task design (top) and trial design (bottom). There were two competition blocks, each containing 20 trials (top). Events and time flow in one trial were also depicted (bottom). (**B**) Optode probe set. The set was placed on the prefrontal and bilateral temporoparietal regions. The integers on the brain surface indicate the recording channels.

**Figure 2 brainsci-13-00600-f002:**
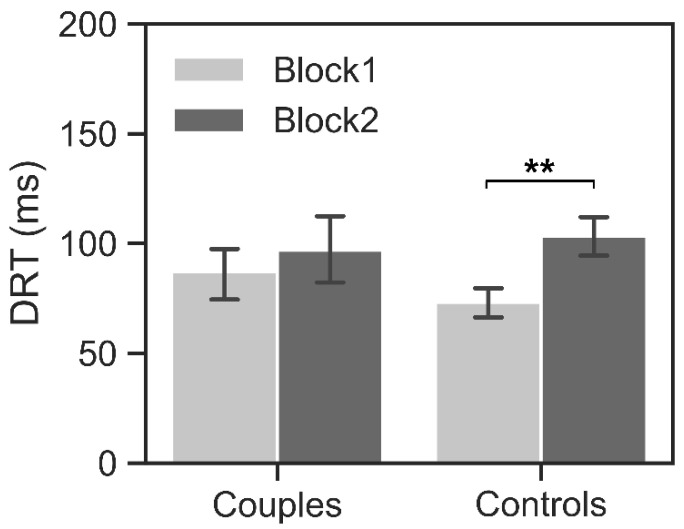
Behavioral performance. The difference in response time (DRT) between the two participants of one pair in block 2 was significantly higher than that in block 1 for controls. ** *p* < 0.01. Error bars indicate one standard error of the means.

**Figure 3 brainsci-13-00600-f003:**
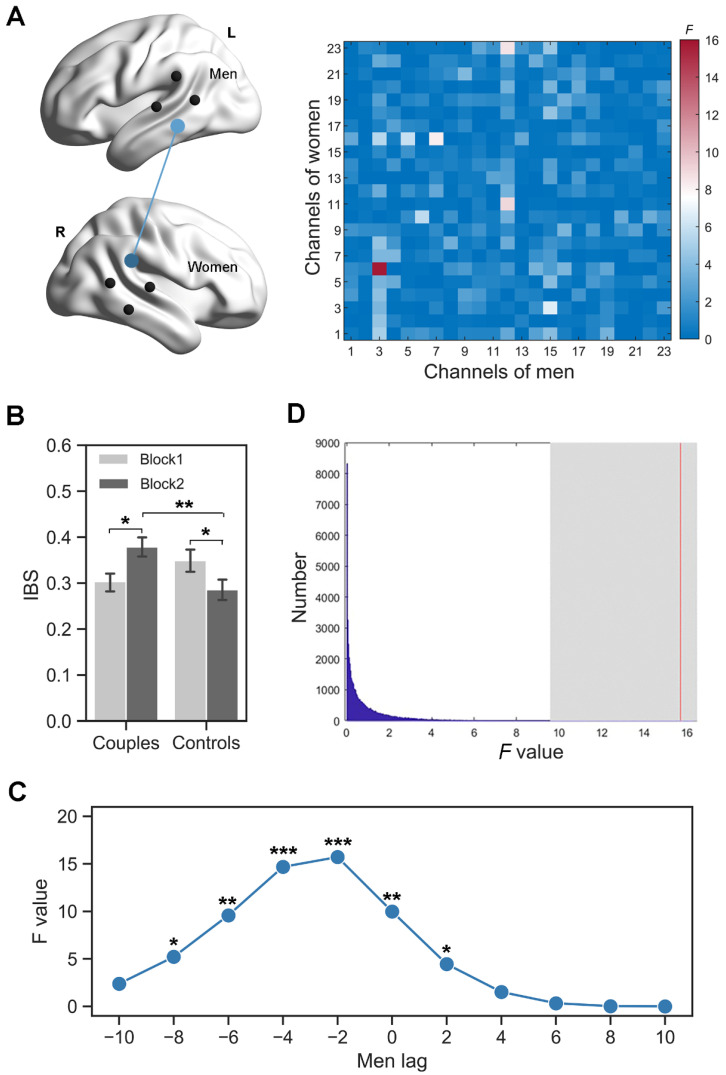
Inter-brain synchronization (IBS). (**A**) The interaction effects (*F* value) between group and task block for all channel combinations (right). Black dots indicate the position of optode probes (left). (**B**) Comparisons of the MTC-TPJ IBS. (**C**) A significant interaction effect between group and task block across several time-lags at the MTC–TPJ IBS. (**D**) The permutation test. The red line indicates the positions of the interaction effect for the real pairs relative to the distribution of the permutated data. * *p* < 0.05, ** *p* < 0.01, *** *p* < 0.001. Error bars indicate one standard error of the means.

**Figure 4 brainsci-13-00600-f004:**
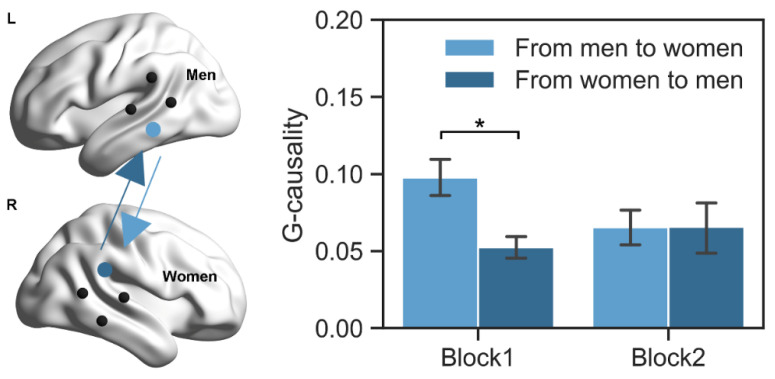
Directional coupling in couples. The G-causality was analyzed in two directions (from men to women, from women to men). Couples showed greater G-causality from men to women (vs. from women to men) in block 1, but no significant difference between the two directions in block 2. * *p* < 0.05. Black dots indicate the position of optode probes (**left**). Error bars indicate one standard error of the means (**right**).

**Figure 5 brainsci-13-00600-f005:**
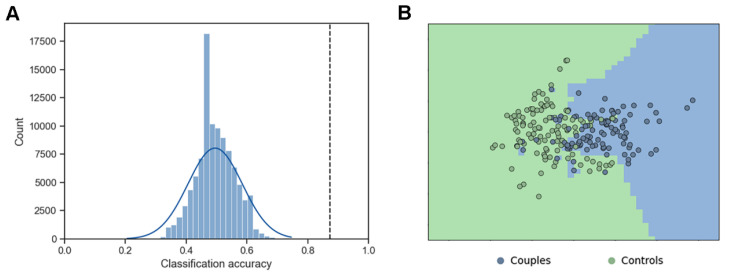
Classification results. (**A**) The classification accuracy of couple and controls. The dashed line indicates the positions of the classification accuracy for the original data. (**B**) Scatter plots depicted subjects plotted on a 2D plane using multidimensional scaling. Green and blue background colors presented areas where pairs of participants were classified as couples and controls, respectively, using a KNN classifier that was trained on the whole group. The data were plotted just for visualization and were not subjected to further statistical testing.

## Data Availability

The data that support the findings of this study are available from the corresponding author upon reasonable request.

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
