# Peer review of "Interpersonal Competition in Elderly Couples: A Functional Near-Infrared Spectroscopy Hyperscanning Study"

_brainsci, 2023, doi:10.3390/brainsci13040600_

Round 1

Reviewer 1 Report

The authors attempt to address the question: what are the potential mechanisms behind interpersonal competition in elderly people?  The authors designed a competitive button-pressing task and collected brain signals through fNIRS hyperscanning data, and found that changes in inter-brain synchronization may explain maintenence of interpersonal competition. 

Overall, I think the conclusion requires more controls. Despite that the authors used matched-cross-sex elderly as control group, I think a proper control group should also include factors like depression, life habits, overall cognitive functions, motor control abilities. 

Reviewer 2 Report

Please revise the paper by addressing the following issues:

1.      Abstract should contain the summary of whole research.

2.      Problem statement and research gap should be precisely highlighted.

3.      Would you please reason both the novelty and the relevance of your paper goals?

4.      “Methods” should be “Material and Methods”.

5.      The most important element of research paper i.e. Conclusion is missing. The conclusion is intended to help the reader understand why your research should matter to them after they have finished reading the paper. It offers new insight and creative approaches for framing/contextualizing the research problem based on the results of your study.

6.      Please include some practical implications of your study findings in the conclusion.

7.      Limitations and future research should be discussed under separate heading after conclusion.

8.      The captions of the Figures should be concise and extra detail should be provided in the main text.  

9.      References require some revisions for uniformity in pattern according to the style recommended by the Journal.

Please proof read the manuscript before submitting the revision. 

Reviewer 3 Report

It is a very original and interesting study. 

However, some clarifications must be added in the method section.

1°) You said that five of the subjects failed for technical reasons.

Clarify which one please.

2°) are you sure that you avoided the Mayer waves are cyclic changes or waves in arterial blood pressure brought about by oscillations in baroreceptor and chemoreceptor reflex control systems. The waves are seen both in the ECG and in continuous blood pressure curves and have a frequency about 0.1 Hz (10-second waves). Your full frequency range 0.01-1 HZ are closed to them (0.11-0.14 HZ)

3°) You did not analyzed the HbR and preferred to apply PCA , DWT and DS according your excellent prior works (2005) . Explain why please.

Conclusion

I am afraid that in reality you measure the stress that is higher in the coupe who do not know them selves before via the parasympathic activity via the Mayer waves.

Maybe you have to add this point in the discussion and limit of the study ?
